# Concurrent Use of Renal Replacement Therapy during Extracorporeal Membrane Oxygenation Support: A Systematic Review and Meta-Analysis

**DOI:** 10.3390/jcm10020241

**Published:** 2021-01-11

**Authors:** Saikat Mitra, Ryan Ruiyang Ling, Chuen Seng Tan, Kiran Shekar, Graeme MacLaren, Kollengode Ramanathan

**Affiliations:** 1Cardiothoracic Intensive Care Unit, National University Heart Centre, National University Hospital, Singapore 119228, Singapore; graeme_maclaren@nuhs.edu.sg (G.M.); ram_ramanathan@nuhs.edu.sg (K.R.); 2Yong Loo Lin School of Medicine, National University of Singapore, Singapore 119077, Singapore; ryan.ling@u.nus.edu; 3Saw Swee Hock School of Public Health, National University of Singapore, Singapore 119077, Singapore; ephtcs@nus.edu.sg; 4Adult Intensive Care Services, The Prince Charles Hospital, Brisbane, QLD 4032, Australia; kiran.shekar@health.qld.gov.au; 5Institute of Health and Biomedical Innovation, Queensland University of Technology, Brisbane, QLD 4000, Australia; 6Faculty of Medicine, University of Queensland, Brisbane, QLD 4072, Australia; 7Faculty of Medicine, Bond University, Gold Coast, QLD 4226, Australia

**Keywords:** extracorporeal membrane oxygenation, renal replacement therapy, acute kidney injury, mortality

## Abstract

Patients supported with extracorporeal membrane oxygenation (ECMO) often receive renal replacement therapy (RRT). We conducted this systematic review and meta-analysis (between January 2000 and September 2020) to assess outcomes in patients who received RRT on ECMO. Random-effects meta-analyses were performed using R 3.6.1 and certainty of evidence was rated using the Grading of Recommendations, Assessment, Development, and Evaluation (GRADE) approach. The primary outcome was pooled mortality. The duration of ECMO support and ICU/hospital lengths of stay were also investigated. Meta-regression analyses identified factors associated with mortality. A total of 5896 adult patients (from 24 observational studies and 1 randomised controlled trial) were included in this review. Overall pooled mortality due to concurrent use of RRT while on ECMO from observational studies was 63.0% (95% CI: 56.0–69.6%). In patients receiving RRT, mortality decreased by 20% in the last five years; the mean duration of ECMO support and ICU and hospital lengths of stay were 9.33 days (95% CI: 7.74–10.92), 15.76 days (95% CI: 12.83–18.69) and 28.47 days (95% CI: 22.13–34.81), respectively, with an 81% increased risk of death (RR: 1.81, 95% CI: 1.56–2.08, *p* < 0.001). RRT on ECMO was associated with higher mortality rates and a longer ICU/hospital stay compared to those without RRT. Future research should focus on minimizing renal dysfunction in ECMO patients and define the optimal timing of RRT initiation.

## 1. Introduction

Almost 50% of patients on extracorporeal membrane oxygenation (ECMO) require renal replacement therapy (RRT) [1]. The indications for initiating RRT while on ECMO are similar to other critically ill patients and can be multifactorial [1,2]. Acute kidney injury (AKI) is a common indication in almost 80% of patients receiving extracorporeal membrane oxygenation (ECMO) [3]. The aetiology of AKI in the ECMO patient population can be attributed to pre-ECMO and ECMO factors such as hypoxaemia and haemodynamic perturbations around the time of initiation, low cardiac output state, severe right heart dysfunction, underlying multisystem disorders, systemic inflammation, hormonal imbalances, exposure to nephrotoxins, and ischaemic-reperfusion injury [2,3,4,5,6,7]. The mortality of critically ill patients who develop AKI is estimated to be 40–70% [8,9]. On the other hand, the reported incidence of mortality due to AKI associated with ECMO is approximately 80% [6,10,11,12].

Patients needing ECMO have fluid and electrolyte imbalances which can be regulated better using RRT. Fluid balance on day 3 of ECMO has been found to be an independent marker of mortality in some studies and the use of RRT to offset fluid overload has been shown to improve clinical outcomes [13]. RRT also helps clear dialyzable toxins where ECMO would be needed for hemodynamic stability [14,15]. Nonetheless, previous reviews have shown that the use of RRT on ECMO is associated with increased mortality in both adult and paediatric patients [16,17]. We performed a systematic review and meta-analysis to examine how the use of RRT may affect outcomes in adult patients receiving ECMO.

## 2. Materials and Methods

A systematic search was conducted after registering on the International Prospective Register of Systematic Reviews (PROSPERO CRD42020188331). The review of literature followed the Preferred Reporting Items for Systematic Reviews and Meta-Analysis (PRISMA) statement in three major international medical bibliographical databases (PubMed, EMBASE, and Cochrane) from 1st January 2000 to 30th September 2020. The search strings included the Boolean terms “AND”, “OR” and “NOT” with the following keywords and their respective variants or derivatives in any relevant combination: renal replacement therapy, haemofiltration, haemodiafiltration, continuous renal replacement therapy, continuous venovenous haemodialysis, continuous venovenous haemodiafiltration, continuous venovenous haemofiltration, continuous arteriovenous haemodialysis, and extracorporeal membrane oxygenation. We have included randomised controlled trials, case-control studies, cohort studies and case series (sample size of minimum 10 patients). We also included studies where patients were on RRT prior to initiation of ECMO. Studies related to animals, paediatric patients (<18 years), pregnant patients, pharmacokinetics, technical aspects of ECMO and RRT, studies involving mechanical circulatory support other than ECMO, those published from the same centres and covering the same time period as well as those published in non-English languages were excluded. Publications reporting on Extracorporeal Life Support Organisation (ELSO) registry data were also excluded to avoid duplication of data. Additionally, we considered national databases rather than single centre data where applicable to avoid overlapping.

A hand search of all relevant studies and their citation lists was performed to identify articles for inclusion. The eligibility of the studies was independently assessed by two reviewers (SM and RRL) and any conflicts were resolved by consensus or by a third reviewer (KR). Included studies were reviewed using the appropriate Joanna Briggs Institute (JBI) checklists.

The following data were extracted for each trial: study design (duration of study, type of study, country of origin of study centre, year of publication), patient demographics (sample size, number of patients on RRT, number of male/female patients, mean age), pre-RRT characteristics (indications for ECMO, cannulation strategy (veno-venous [VV] or veno-arterial [VA] ECMO), pre-RRT serum lactate and creatinine), number of patients with other multi-organ failures (MOFs) apart from the primary organ failure for which ECMO was initiated (including liver failure, bowel ischaemia, acute stroke, intracranial haemorrhage and disseminated intravascular coagulation), number of patients with new-onset infections (sepsis or bacteraemia) after ECMO initiation and relevant clinical outcomes (mortality, hospital and intensive care unit (ICU) length of stay, and ECMO duration).

### Statistical Analysis

Our primary outcome was overall mortality due to concurrent use of RRT while on ECMO. Overall mortality for this review was defined as in-hospital mortality, ICU mortality, 30-day mortality or 90-day mortality. Secondary outcomes included the mean duration of ECMO support and ICU and hospital lengths of stay in patients with combined therapies. Additionally, we measured the pooled incidence of other MOFs and new-onset infections in patients who were on both ECMO and RRT therapies compared to those who were treated with ECMO alone.

As a high degree of inter-study heterogeneity was expected, random effects meta-analyses (DerSimonian and Laird) [18,19] were conducted on R 3.6.1 using the meta (v4.12-0) and dmetar (v0.0.9000) packages, and confidence intervals (CI) were computed using the Clopper–Pearson method [20]. Mortality outcomes are presented as pooled proportions and 95% confidence intervals (CI), and dichotomous outcomes are presented as odds ratios (ORs) and 95% CI. Planned subgroup analyses were conducted with continuity correction to allow the inclusion of studies with zero events and included the other reported mortalities (in-hospital, ICU, 30-day or 90-day), the geographical location (Asia, Europe, America, and Australia), the presence of renal replacement therapy (RRT and no RRT), the duration ECMO (more and less than 7 days) and the year of publication (before and after 2016). Summary-level meta-regression was conducted if a minimum of 6 data points could be collected to explore potential sources of heterogeneity or prognostically-relevant study-level covariates [18]. We used the Grading of Recommendations, Assessment, Development, and Evaluation (GRADE) guidance to assess between-study heterogeneity and rated the certainty of evidence using the GRADE approach [21,22,23]. We used the ‘GRADEpro’ app to rate the evidence [24] and presented in GRADE evidence profiles and summaries of findings tables using standardised terms [25,26].

Publication bias was assessed using Egger’s test. Leave-one-out sensitivity analysis (LOO) was performed for all analyses by omitting 1 study at a time to identify outliers or influential studies. Means and standard deviations of continuous variables were pooled using the methods proposed by Wan et al. [27].

## 3. Results

Our preliminary search identified 2343 articles and 404 duplicates were removed. Of the 1939 articles screened, 1769 studies were excluded after examining the abstract. We obtained 155 citations in full text and 130 of these studies did not meet our inclusion criteria (Appendix A: PRISMA diagram). In total, 25 studies detailing 5896 adult patients that reported on the use of RRT and ECMO were included (Table 1) [28,29,30,31,32,33,34,35,36,37,38,39,40,41,42,43,44,45,46,47,48,49,50,51,52]. Twenty-four studies were observational in nature [28,29,30,31,32,33,34,35,36,37,38,39,40,41,42,43,44,45,46,47,48,49,50,51], while one study was a randomised controlled trial (RCT) [52]. For our quantitative analysis, we included 24 observational studies (5855 patients), while the findings of the RCT were reported separately. Overall, 3223 patients received combined therapy with ECMO and RRT (both observational studies and RCT). The quality assessment of the studies was performed using the JBI checklists (Appendix A), which revealed that the studies were of the highest quality. Continuous renal replacement therapy (CRRT) was the most commonly used modality in these patients.

### 3.1. Demographic Analysis

#### 3.1.1. Observational Studies

The pooled mean age of patients (Appendix A) receiving RRT on ECMO was 50.9 years (95% CI: 46.9–54.8). The proportion of male patients across the studies (Appendix A) was 68.2% (95% CI: 64.4–71.9%). After removing the two outliers detected by LOO analysis [29,31], the proportion of males was 67.0% (95% CI: 63.0–70.8%). Pooled proportion of concurrent use of VA-ECMO with RRT from 22 studies (Appendix A) was 71.8% (95% CI: 49.8–89.6%) with a significant publication bias (*P_egger_* = 0.007). Pooled mean serum lactate at the initiation of combined therapies from six studies was 3.79 mmol/L (95% CI: 2.41–5.17). After the removal of one outlier [43], the pooled serum lactate level was estimated to be 4.22 mmol/L (95% CI: 3.29–5.15). The pooled mean serum creatinine at the initiation of RRT from four studies was calculated to be 2.12 mg/dL (95% CI: 1.75–2.49). After removing the only outlier detected by LOO [29], the pooled serum creatinine was calculated to be 2.25 mg/dL (95% CI: 1.84–2.66).

#### 3.1.2. RCT

The mean age of the patients receiving both the therapies in the study by Li et al. [52] was 61.2 ± 8.3 years and all of them were on VA-ECMO. Around 72.2% were males who required combined therapies. Initial serum lactate and creatinine levels were 12.5 ± 8.3 mmol/L and 1.1 ± 0.4 mg/dL, respectively.

### 3.2. Primary Outcome

#### 3.2.1. Observational Studies

The pooled overall mortality (Figure 1) in patients due to the use of RRT on ECMO (24 studies) was 62.9% (95% CI: 56.0–69.6%). Subgroup analysis found no significant differences (*p* = 0.57) between in-hospital mortality (14 studies, 60.1%, 95% CI: 50.1–69.8%) [31,33,34,35,36,37,38,39,42,43,45,48,49,50], ICU mortality (3 studies, 67.3%, 95% CI: 60.1–74.1%) [28,29,32], 30-day mortality (3 studies, 68.4%, 95% CI: 58.0–77.9%) [41,44,47], and 90-day mortality (3 studies, 57.0%, 95% CI: 17.6–92.0%) [30,40,46]. Only one study mentioned overall mortality [51].

Subgroup analysis found that mortality was significantly different when considering the publication years (before and after 2016), the presence of RRT, the duration of ECMO, and geographical region. The pooled mortality (Appendix A) prior to 2016 (10 studies) was 74.1% (95% CI: 60.0–86.2%), while pooled mortality after 2016 (14 studies) was 56.1% (95% CI: 47.7–64.5%, *p* = 0.03). The presence of RRT was associated with a significant increase in mortality (19 studies, Relative Risk (RR): 1.81, 95% CI: 1.56–2.08, *p* < 0.001, Figure 2) when compared to patients on ECMO alone. After removing the only outlier detected by LOO, [28] this increased risk of mortality remained significant (RR: 1.84, 95% CI: 1.63–2.08, *p* < 0.001), with significant publication bias (*P_egger_* = 0.02). Additionally, the pooled mortality among patients (Appendix A) with ECMO durations of less than 7 days (3 studies) was 66.5% (95% CI: 54.5–77.5%), compared to 41.5% (95% CI: 33.3–49.9%, *p* < 0.001) for those with ECMO durations of more than 7 days (4 studies). Pooled mortality reported by studies from Asia (11 studies, 65.7%, 95% CI: 54.8–74.0%), Europe (7 studies, 65.7%, 95% CI: 53.9–76.6%), and America (5 studies, 59.4%, 95% CI: 38.6–78.6%) were relatively similar, and higher than those reported from Australia (1 study, 33.0%, 95% CI: 24.2–42.4%).

#### 3.2.2. RCT

Li et al. [52] reported that the patients who received both ECMO and RRT had 61.9% mortality at the end of 30 days.

### 3.3. Secondary Outcomes

#### 3.3.1. Observational Studies

The mean ECMO duration (7 studies), ICU length of stay (3 studies) and hospital length of stay (4 studies) in patients receiving combined therapies were 9.33 days (95% CI: 7.74–10.92), 15.76 days (95% CI: 12.83–18.69) and 28.47 days (95% CI: 22.13–34.81), respectively (Figure 3).

The patients on combined therapies had a higher incidence of both other MOFs {3 studies, 31.9% (95% CI: 17.8–47.7%)} and new-onset infections {2 studies, 17.3% (95% CI: 15.5–19.2%)} in contrast to 15.5% (other MOFs: 95% CI: 10.2–21.5%) and 11.9% (New-onset infections: 95% CI: 5.87–19.5%) in patients on ECMO alone. Patients who received concurrent RRT while on ECMO had a significantly higher incidence of new-onset infections (RR: 1.65, 95% CI: 1.39–1.97, *p* < 0.001); this was not observed for other MOFs (RR: 2.22, 95% CI: 0.93–5.22, *p* = 0.072) in patients who had concurrent use of RRT while on ECMO.

#### 3.3.2. RCT

The median durations of ECMO, ICU length of stay (LOS) and hospital LOS were 110.6 h {Interquartile Range (IQR): 94.6–144.5}, 10.5 days (IQR: 7.0–14.6) and 20.5 days (IQR: 15.8–29.3), respectively, in the combined group. The authors also noted that the incidence of stroke and infection was 9.5% and 33.3%, respectively, in patients who received concurrent RRT while on ECMO.

### 3.4. Meta-Regression Analysis

Meta-regression analysis of observational studies (Table 2) showed that longer ECMO durations were associated with lower odds of mortality (Odds Ratio (OR): 0.97, 95% CI: 0.95–0.98, *p* <0.001), while the need for RRT on VA-ECMO was associated with increased odds of mortality (OR: 1.23, 95% CI: 1.04–1.46, *p* = 0.02). Age, male gender, lactate level and sample size were not predictive of mortality.

### 3.5. Pre-ECMO Vs. Post-ECMO RRT

Three studies reported on the timing of RRT. Deatrick et al. [34] found no significant difference (*p* = 0.19) in terms of survival between the patients who received RRT before ECMO (53%) and after ECMO (36%). On the other hand, Haneya et al. [38] noted that those patients who were on RRT prior to ECMO had a higher mortality rate of 43.4% compared to the survivors (21.2%). Similarly, Panholzer et al. [45] noted that those who were on RRT before ECMO initiation had a mortality of 37.5%.

### 3.6. Risk of Bias

We assessed the certainty of evidence for all of our primary and secondary outcome measures using the GRADE approach (Table 3). The certainty of evidence was high for the mortality of patients who received both therapies compared to those who were treated with ECMO alone, and there was a low certainty of evidence for ICU and hospital LOS.

## 4. Discussion

This review reports pooled mortality outcomes in a heterogenous group of patients who received both ECMO and RRT. A large proportion (72%) of patients included in this review received VA-ECMO. We observed that the most commonly used RRT modality along with ECMO was CRRT; patients were predominantly middle-aged males with a pooled mortality of approximately 63%. This correlated with the mortality reported from the only RCT that reported on the use of RRT during ECMO. The mortality in patients receiving both ECMO and RRT has decreased significantly in last 5 years (20%) compared with that reported till the end of 2015. Most of the studies were observational in nature, while there was one RCT from Asia. The combined use of ECMO and RRT was associated with increased death risk by almost 81% when compared to patients receiving ECMO alone.

All patients on ECMO are at increased risk of inflammatory and haemodynamic perturbations that put them at an increased propensity of developing multiple organ dysfunction [3]. Equally, although non-pulsatile flow generated by VA-ECMO to maintain end-organ perfusion has been postulated to increase the risks of AKI [2,53], there is no robust clinical data to support this hypothesis. In addition, the higher odds of mortality seen in patients receiving both VA-ECMO and RRT in this study may also be attributed to patient selection issues and the timing of ECMO initiation. The pilot RCT conducted by Li et al. [52] in post-cardiotomy VA-ECMO patients showed that early use of RRT in these patients was associated with less mortality compared to those who received RRT late, as per conventional indications. The ELSO registry reports a mortality of 56% in a heterogeneous group of patients supported with VA-ECMO, which includes both patients who did or did not receive RRT support [54]. Given that critically ill patients needing ECMO are sicker, it is plausible that the mortality of the combined extracorporeal therapies would be higher. It can be expected that outcomes with combined VA-ECMO and RRT use may be better in potentially reversible conditions such as myocarditis and certain cardiotoxic drug ingestion (e.g., aluminium phosphide), in which VA-ECMO survival in excess of 60% has been reported [55,56]. We also noted a considerably higher incidence of other organ failures (~32%) and infections (~17%) in the combined group, which could have led to higher mortality in this group.

We observed that shorter ECMO duration was associated with higher mortality in this cohort of patients and vice-versa. Further additional analysis on mortality based on ECMO duration (more and less than 7 days) revealed that mortality was higher in patients with shorter ECMO duration (66.5% vs. 41.5%). This goes in hand with our meta-regression analysis. While there were only a few studies in this analysis, we believe that patients who had a shorter duration of ECMO were sicker and had a higher MOF needing RRT in addition to ECMO, resulting in a higher mortality. The association between longer ECMO duration and lower mortality can be attributed to immortal bias: patients must first survive long enough in order to be weaned off ECMO [57]. A proportion of ECMO patients are likely to die early, either due to progressive multiple organ failure, fatal complications, limited cardiopulmonary recovery, lack of viable bridging options such as transplantation, or simply palliation based on clinician’s judgment or patient and family wishes. There are significant differences between patients with severe respiratory failure and refractory heart failure receiving ECMO. Apart from obvious pathophysiologic differences, achieving sufficient reversibility of underlying pathology to wean from ECMO and survive to hospital discharge is an important consideration. Previous reviews also concluded high mortality in patients who received combined extracorporeal therapies [16,17,58]. Similarly, the mechanisms behind decreasing mortality trends in patients who received both ECMO and RRT over last 5 years could not be understood within the scope of this review. It is possible that better patient selection, timing and improving clinical application played a role in addition to technological advances. Additionally, the number of publications on RRT and ECMO has increased in last 5 years, which has resulted in more granularity in the overall data. Whether ECMO is a risk factor for AKI or whether early ECMO mitigates the development of AKI and other organ failures remains a much-debated entity, given the higher mortality reported in this cohort.

Our systematic review has several limitations. This analysis is based mainly on observational studies with significant heterogeneity. The random-effects model was used when conducting this meta-analysis for the anticipated heterogeneity in addition to using the GRADE approach to rate the certainty of evidence. We performed additional subgroup analysis to account for heterogeneity. Furthermore, meta-regression analyses are constrained by an inherent lack of power and increase the risk of Type II errors. The GRADE assessment showed low to high levels of certainty for the results of the analysis. An increased ECMO duration was associated with less odds of mortality in patients receiving combined supports in this review. Finally, the Egger’s test yielded non-significant results for most of our primary endpoints, except for relative risk of mortality due to combined therapies which had significant publication bias. Nonetheless, JBI appraisal of the included studies suggests that they were of high quality, limiting the possibility of publication bias. Few studies assessed the incidence of other MOFs or new-onset infections, so our results should be interpreted with caution and may be considered hypothesis-generating. We did not analyse any differences in outcomes based on timing of RRT initiation, different modalities of RRT, different ventricular unloading techniques while on ECMO, or different forms of shock because very few studies examined these data.

## 5. Conclusions

Adult patients receiving both ECMO and RRT are at a greater risk of death. The mortality, however, has shown a decreasing trend over the last 5 years. Patients receiving RRT on VA-ECMO have greater odds of death compared with those receiving RRT on VV-ECMO. Given the higher mortality and morbidity in the group of patients who received RRT on ECMO, future research should focus on determining the optimal timing of both VA and VV ECMO initiation in order to potentially mitigate AKI. Further studies should also explore the optimal timing of RRT initiation during ECMO in patients with appropriate indications.

## Figures and Tables

**Figure 1 jcm-10-00241-f001:**
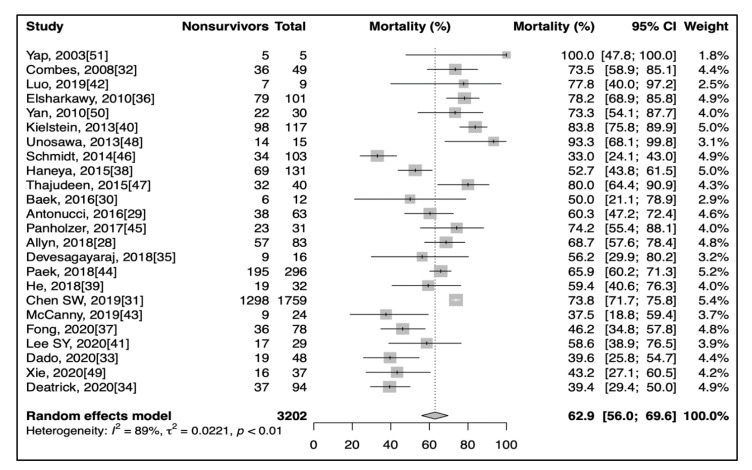
Forest plot showing pooled mortality.

**Figure 2 jcm-10-00241-f002:**
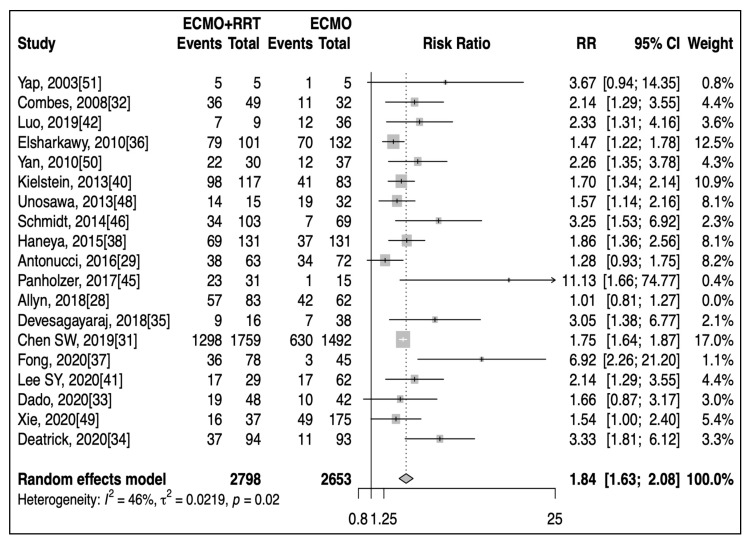
Forest plot showing increased risk of mortality in patients receiving combined therapies (ECMO: extracorporeal membrane oxygenation, RRT: renal replacement therapy).

**Figure 3 jcm-10-00241-f003:**
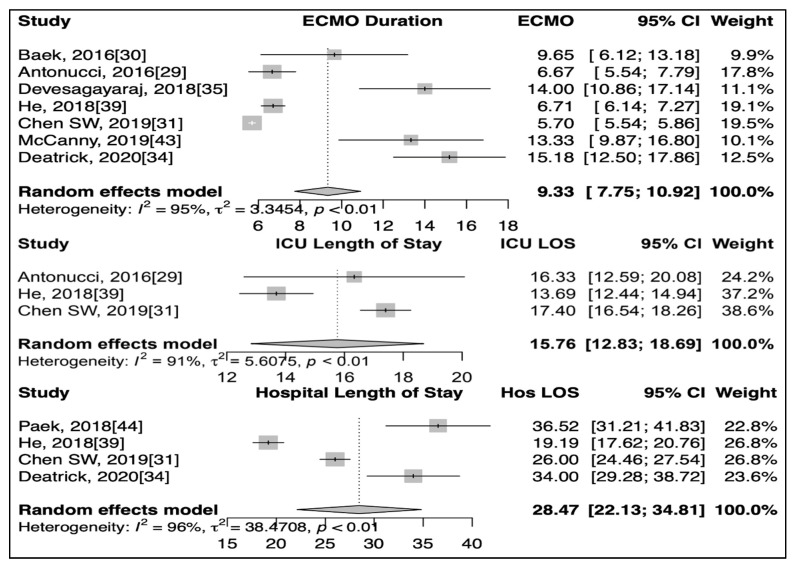
Forest plot showing mean ECMO (extra-corporeal membrane oxygenation) duration, ICU (Intensive care unit) and Hospital Length of Stay (LOS) of patients who received combined therapies.

**Table 1 jcm-10-00241-t001:** Summary of all studies.

				Outcome		
Author/Year	Study Type, Sample Size	Country	ECMO/RRT	Mortality	ICU/Hospital LOS	ECMO Duration
Allyn, 2018 [28]	Cohort, 145	France	+	+	−/−	−
Antonucci, 2016 [29]	Cohort, 135	Belgium	+	+	+/−	+
Baek, 2016 [30]	Cohort, 12	Korea	+	+	−/−	+
Chen, 2019 [31]	Cohort, 3251	Taiwan	+	+	+/+	+
Combes, 2008 [32]	Cohort, 81	France	+	+	−/−	−
Dado, 2020 [33]	Cohort, 90	USA	+	+	−/−	−
Deatrick, 2020 [34]	Cohort, 187	USA	+	+	−/+	+
Devasagayaraj, 2018 [35]	Cohort, 54	USA	+	+	−/−	+
Elsharkawy, 2010 [36]	Cohort, 233	USA	+	+	−/−	−
Fong, 2020 [37]	Cohort, 123	HK SAR	+	+	−/−	−
Haneya, 2015 [38]	Cohort, 262	Germany	+	+	−/−	−
He, 2018 [39]	Cohort, 32	China	+	+	+/+	+
Kielstein, 2013 [40]	Cohort, 200	Germany	+	+	−/−	−
Lee SY, 2020 [41]	Cohort, 91	Korea	+	+	−/−	−
Luo, 2009 [42]	Cohort, 45	China	+	+	−/−	−
McCanny, 2019 [43]	Cohort, 24	Ireland	+	+	−/−	+
Paek, 2018 [44]	Cohort, 296	Korea	+	+	−/+	−
Panholzer, 2017 [45]^.^	Cohort, 46	Germany	+	+	−/−	−
Schmidt, 2014 [46]	Cohort, 172	Australia	+	+	−/−	−
Thajudeen, 2015 [47]	Cohort, 40	USA	+	+	−/−	−
Unosawa, 2013 [48]	Cohort, 47	Japan	+	+	−/−	−
Xie, 2020 [49]	Cohort, 212	China	+	+	−/−	−
Yan, 2010 [50]	Cohort, 67	China	+	+	−/−	−
Yap, 2003 [51]	Case-control, 10	Taiwan	+	+	−/−	−
Li, 2019 [52]	RCT, 41	China	+	+	+/+	+

ECMO: extracorporeal membrane oxygenation, RRT: renal replacement therapy, ICU: intensive care unit, LOS: length of stay.

**Table 2 jcm-10-00241-t002:** Meta-regression analysis of covariates.

Covariates	Studies	Odds Ratio	Lower CI	Upper CI	*p* Value
VA-ECMO	24	1.23	1.04	1.46	**0.02**
ECMO duration	7	0.97	0.95	0.98	**<0.001**
Lactate	7	1.08	0.95	1.24	0.22
Age	9	1.01	0.99	1.03	0.35
Male	10	0.82	0.17	3.94	0.81
Sample size	24	1.00	1.00	1.00	0.73

**Table 3 jcm-10-00241-t003:** Grading of Recommendations, Assessment, Development, and Evaluation (GRADE) findings.

No. of Studies	Certainty Assessment						Effect			Certainty	Importance
	Study Design	Risk of Bias	Inconsistency	Indirectness	Imprecision	Other Considerations	No. of Events	No. of Individuals	Rate (95% CI)		
**Mortality between patients supported with concurrent ECMO and RRT**											
24	observational studies	not serious	not serious ^a^	not serious	not serious ^b^	none	2175	3202	63.0% (56.0% to 69.6%)	**⊕⊕⊕⊕** **HIGH**	CRITICAL
**ICU Length of Stay**											
3	observational studies	not serious	serious ^c^	not serious	serious ^d^	none	-	1854	15.76 days (12.83 to 18.69)	**⊕⊕** **◯◯** **LOW**	IMPORTANT
**Hospital Length of Stay**											
4	observational studies	not serious	serious ^c^	not serious	serious ^d^^,e^	none	-	2181	29.00 days (21.74 to 36.26)	**⊕⊕** **◯◯** **LOW**	IMPORTANT

**Explanations:**^a^ There was considerable heterogeneity (I^2^ = 89.4%). However, subgroup analysis by geographical region, ECMO duration and time period found significant differences between subgroups among patients. Furthermore, meta-regression found that ECMO duration was significantly associated with increased survival, and VA ECMO with decreased survival. Visual inspection of the forest plots found that there was some variability in the point estimates, but the 95% CIs mostly overlapped. ^b^ The 95% CI are relatively narrow compared to the pooled estimate. In addition, there is a relatively large sample size of 3202 patients, which would reduce imprecision. ^c^ There was considerable heterogeneity. Further to this, visual inspection of the forest plot showed that the point estimates were sparsely distributed. ^d^ Very few studies reported on the outcome, yielding a small sample size that hampers precision. ^e^ The 95% CI is relatively wide in relation to the pooled estimate. **High:** We are very confident that the true prognosis (probability of future events) lies close to that of the estimate [23], **Moderate:** We are moderately confident that the true prognosis (probability of future events) is likely to be close to the estimate, but there is a possibility that it is substantially different [23], **Low:** Our confidence in the estimate is limited: the true prognosis (probability of future events) may be substantially different from the estimate [23], **Very low:** We have very little confidence in the estimate: the true prognosis (probability of future events) is likely to be substantially different from the estimate [23].

## Data Availability

Not applicable.

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
