# Peer review of "Concurrent Use of Renal Replacement Therapy during Extracorporeal Membrane Oxygenation Support: A Systematic Review and Meta-Analysis"

_jcm, 2021, doi:10.3390/jcm10020241_

Round 1

Reviewer 1 Report

The authors have performed a meta-analysis of studies dealing with renal replacement therapy in patients who received ECMO therapy. The results of the studies indicate that the mortality of patients concurrent use with these two methods is extremely high. The study is well methodologically structured, correctly written. I have only one minor remark.

Line 95: What does statement "duration of mortality"  mean? I am not sure this is correct.

Author Response

Reviewer 1 report:

The authors have performed a meta-analysis of studies dealing with renal replacement therapy in patients who received ECMO therapy. The results of the studies indicate that the mortality of patients’ concurrent use with these two methods is extremely high. The study is well methodologically structured, correctly written.

Response 1: We thank the reviewer for kind appreciation.

I have only one minor remark.

Line 95: What does statement "duration of mortality" mean? I am not sure this is correct.

Response 2: We thank the reviewer for the constructive comment. We have changed the phrase to ‘other reported mortalities’ to denote we have done subgroup analysis to find any significant difference between in-hospital, ICU, 30-day or 90-day mortalities (line no. 100).

Reviewer 2 Report

In my review of the article titled "CONCURRENT USE OF RENAL REPLACEMENT THERAPY DURING EXTRACORPOREAL MEMBRANE OXYGENATION SUPPORT: A SYSTEMATIC REVIEW AND META-ANALYSIS", the authors describe their analysis and literature review of combined use of Renal Replacement Therapy (RRT) and Extracorporeal Membrane Oxygenation (ECMO) support. The authors looked at the outcomes of combined RRT/ECMO in 24 observational studies and 1 randomized trial. This is a well-written manuscript, addressing an important clinical question. My comments are below.

1- This is a retrospective analysis of published studies looking at the pooled mortality of RRT/ECMO compared to ECMO alone. As expected, patients who had combined RRT/ECMO had higher mortality compared to ECMO alone. AKI itself is associated with higher risk of mortality in shock patients. In addition, this could be potentially explained by the fact that these patients requiring RRT likely have multi-organ failure in the setting of hypo-perfusion and shock. Did authors look at the incidence of multi-organ failure (including liver and respiratory failure, etc) in these patients? 

2- Authors describe that duration of ECMO was associated with lower odds for mortality with a significant p value. Can authors be more clear on whether "longer versus shorter duration of ECMO" was associated with lower odds of mortality. Can authors further explain why they think they found this in their analysis. 

3- Authors explained potential hemodynamic changes that RRT might alter in those patients on ECMO, including optimizing volume status and pre-load. Did authors find any other measures used to unload the ventricles while on ECMO (ie use of Impella) in these studies? Could authors look at the correlation between these other measures to unload the ventricle and the outcomes/pooled mortality in this analysis? If this is not possible, authors should include this in the limitation section.

4- Did authors see a difference in the outcomes based on the type of shock, cardiogenic versus non-cardiogenic (septic.. etc)? If this is not possible, authors should include this point in the limitation section.

Author Response

Reviewer 2 report:

In my review of the article titled "CONCURRENT USE OF RENAL REPLACEMENT THERAPY DURING EXTRACORPOREAL MEMBRANE OXYGENATION SUPPORT: A SYSTEMATIC REVIEW AND META-ANALYSIS", the authors describe their analysis and literature review of combined use of Renal Replacement Therapy (RRT) and Extracorporeal Membrane Oxygenation (ECMO) support. The authors looked at the outcomes of combined RRT/ECMO in 24 observational studies and 1 randomized trial. This is a well-written manuscript, addressing an important clinical question.

Response 1: We thank the reviewer for the kind acknowledgement.

My comments are below.

1- This is a retrospective analysis of published studies looking at the pooled mortality of RRT/ECMO compared to ECMO alone. As expected, patients who had combined RRT/ECMO had higher mortality compared to ECMO alone. AKI itself is associated with higher risk of mortality in shock patients. In addition, this could be potentially explained by the fact that these patients requiring RRT likely have multi-organ failure in the setting of hypo-perfusion and shock. Did authors look at the incidence of multi-organ failure (including liver and respiratory failure, etc) in these patients?

Response 2: We thank the reviewer for the constructive comments. Further to reviewer’s suggestion, we checked the shortlisted articles as well as our data extraction sheet again and re-analysed the patient data on other organ failures and new-onset infections (line no. 82 – 85, 91 - 93). We noted that the incidence of other organ failures (31.9%) and new-onset infections (17.3%) were quite high in the combined group compared to the group on ECMO without RRT (organ failures: 15.5% and infections: 11.9%); and higher mortality in combined group may be attributable to this. We have included this in our result section (line no. 182 - 188, 192 - 193) and discussion section as well (line no. 240 - 242). As the studies are less in number, this should be interpreted cautiously and included as limitation (line no. 275 - 277).

2- Authors describe that duration of ECMO was associated with lower odds for mortality with a significant p value. Can authors be more clear on whether "longer versus shorter duration of ECMO" was associated with lower odds of mortality. Can authors further explain why they think they found this in their analysis.

Response 3: We thank the reviewer for the interesting question. Our meta-regression showed that a shorter duration of ECMO was associated with a higher mortality and vice versa (line no. 195 - 196). While the association of mortality with longer duration of ECMO was explained by immortal time bias, we did further subgroup analysis to look at the impact of shorter duration of ECMO and mortality. Additional subgroup analysis on mortality based (line no. 102, 156) on ECMO duration (more and less than 7 days) showed that mortality was higher in patients who had ECMO duration less than 7 days (66.5% vs 41.5%, line no. 163 - 165). While the number of studies that highlighted this was less in number, this is hypothesis generating and we believe that patients who had shorter duration of ECMO were a lot sicker and had a higher MOF needing RRT in addition to ECMO, resulting in a higher mortality (line no. 243 - 250).

3- Authors explained potential hemodynamic changes that RRT might alter in those patients on ECMO, including optimizing volume status and pre-load. Did authors find any other measures used to unload the ventricles while on ECMO (i.e., use of Impella) in these studies? Could authors look at the correlation between these other measures to unload the ventricle and the outcomes/pooled mortality in this analysis? If this is not possible, authors should include this in the limitation section.

Response 4: We extend our gratitude to the reviewer for the constructive comment. We went through all the included articles again; one study each reported on the use of IABP  and LVAD in ECMO + RRT population. Data related to various ventricular unloading techniques being not granular, it is difficult to make further analyses and  an appropriate conclusion. Hence we have added this as a limitation (line no. 277 - 279).

4- Did authors see a difference in the outcomes based on the type of shock, cardiogenic versus non-cardiogenic (septic.. etc)? If this is not possible, authors should include this point in the limitation section.

Response 5: Again, we convey our thanks to the reviewer for the constructive comment. As large proportion (72%) of our population were on VA-ECMO, we speculate that most of them had suffered from cardiogenic shock. Again, due to lack of granularity in data, it’s difficult to analyse the difference in the outcomes based on different types of shock. Hence, we have included this as one of our limitations (line no. 277 - 279).
